# Untargeted Metabolomics Using UHPLC-HRMS Reveals Metabolic Changes of Fresh-Cut Potato during Browning Process

**DOI:** 10.3390/molecules28083375

**Published:** 2023-04-11

**Authors:** Baohong Li, Yingjie Fu, Hui Xi, Shan Liu, Wuduo Zhao, Peng Li, Wu Fan, Dingzhong Wang, Shihao Sun

**Affiliations:** 1Flavor Research Center, Zhengzhou University, Zhengzhou 450001, China; 2The Key Laboratory of Tobacco Flavor Basic Research of CNTC, Zhengzhou Tobacco Research Institute, Zhengzhou 450001, China; 3Center of Advanced Analysis and Gene Sequencing, Zhengzhou University, Zhengzhou 450001, China

**Keywords:** fresh-cut potato, browning, UHPLC-HRMS, untargeted metabolomics, Compound Discoverer 3.3

## Abstract

Surface browning plays a major role in the quality loss of fresh-cut potatoes. Untargeted metabolomics were used to understand the metabolic changes of fresh-cut potato during the browning process. Their metabolites were profiled by ultra-high performance liquid chromatography coupled with high resolution mass spectrometry (UHPLC-HRMS). Data processing and metabolite annotation were completed by Compound Discoverer 3.3 software. Statistical analysis was applied to screen the key metabolites correlating with browning process. Fifteen key metabolites responsible for the browning process were putatively identified. Moreover, after analysis of the metabolic causes of glutamic acid, linolenic acid, glutathione, adenine, 12-OPDA and AMP, we found that the browning process of fresh-cut potatoes was related to the structural dissociation of the membrane, oxidation and reduction reaction and energy shortage. This work provides a reference for further investigation into the mechanism of browning in fresh-cut products.

## 1. Introduction

Potato (*Solanum tuberosum* L.) is one of the most important tubers for human consumption globally, which is largely used for food and provides an excellent source of nutrients and energy [1]. The demand of potato is moving from fresh to processed potato products [2]. Fresh-cut potato as a minimally processed product has become more and more popular because of its advantages of health and convenience [3]. However, the surface browning of fresh-cut potato can occur easily during storage, which decreases color quality and market value [4]. Hence, it is of great significance for industrial potato processing to understand the browning mechanism and find ways to prevent browning.

Browning is a common discoloration phenomenon in fresh-cut vegetables and fruits. In normal plant tissues, according to phenol-phenolase regionalized distribution theory, phenolic substances and polyphenol oxidase (PPO) were mainly found in vacuoles and plastids, respectively [5]. However, once cell integrity was destroyed by fresh-cut processing, PPO came into contact with the phenolic substrate, and they were exposed to air simultaneously, which induced the occurrence of enzymatic oxidation reactions as three elements of oxygen, substrate and enzyme coexisted. So after cutting, phenolic substrates were catalyzed to quinones, which could polymerize rapidly and react with the amino or sulphydryl group of proteins and sugars through a non-enzymatic reaction into dark brown pigments with adverse effects on the appearance and consumer preference of the fresh-cut products [6]. Current research on the browning of fresh-cut potato focuses principally on the anti-browning methods, such as heating [7,8], packaging [9,10] and using anti-browning agents [11,12]. However, the metabolic changes of fresh-cut potato during the browning process have not been explored using a metabolomics approach.

Untargeted metabolomics is an approach of metabolomic analysis aiming to comprehensively detect metabolites with molecular weights below 1000 Da from biological systems and identify the metabolites which could play the most relevant role in various processes [13,14]. Based on ultra-high performance liquid chromatography coupled with high resolution mass spectrometry (UHPLC-HRMS), this approach is emerging as a powerful tool for exploring metabolic changes in samples under different conditions with great potential in the food processing field, especially authenticity, quality and safety [15,16,17]. For example, it was used to reveal metabolic changes in electron beam irradiated shrimp meat during refrigerated storage [18], late blight-infected tomato at different times post infection [19], fresh-cut Chinese water chestnuts at different yellowing stages [20], bio-chemical changes in Laowo ham during ripening [21], black garlic during the aging process [22] and chemical constituent changes in instant dark tea during fermentation by *Euroti-um cristatum* [23]. In recent years, untargeted metabolomics has been used to study the browning reaction of several fruits and vegetables, such as lettuce [24,25,26], eggplant [27,28] and apple [29]. Nevertheless, to our knowledge, untargeted metabolomics has not been used to explain the browning process of fresh-cut potato during storage period.

The aim of this study was to gain insight into the entire metabolic changes occurring after potato cutting. To achieve this goal, the potato was cut into slices and stored in the refrigerator (4 °C) for 0, 2, 4, 6 and 8 days. An untargeted metabolomics approach based on UHPLC-HRMS was performed to determine the metabolites presenting in fresh-cut potato during storage. Differences in metabolites were assessed using Compound Discoverer 3.3 software and statistical analysis. Our results not only discovered the key metabolites responsible for the browning process but also elucidated the causes of metabolic changes in relation to the browning process.

## 2. Results and Discussion

### 2.1. Color Analysis

Color was a critical visual quality indicator for fresh-cut fruits and vegetables. As shown in Figure 1A, the initial color of fresh-cut potato slices was yellow, then gradually deepened with the extension of storage time, and finally turned dark brown at the end of storage. The color variation demonstrated that the browning reaction occurred during the storage period. The lightness (L*), redness (a*) and total color difference (ΔE*) values were used to evaluate the color changes of fresh-cut potato slices [30]. L* values decreased and a* values increased with prolonged storage time, indicating that the lightness became dark and redness became strong, respectively (Figure 1B,C). ΔE* values gradually increased during the whole storage period (Figure 1D). In addition, the highest ΔE* value was observed on 8 days of storage, indicating the maximum color difference generally. These results meant that browning tended to aggravate as the storage time progressed, which may be related to oxidative reactions or pigment accumulation [31].

### 2.2. Metabolites Profiling

In this study, 80% methanol aqueous solution was used as the extraction solvent to comprehensively extract the polar and non-polar metabolites of fresh-cut potatoes. Metabolic profiles of 30 fresh-cut potato samples with different periods were analyzed using UHPLC-HRMS under positive and negative ionization modes. The typical total ion chromatograms (TICs) of fresh-cut potatoes (single 0d sample and 8d sample was referenced as the example) were shown in Figure 2. After peak alignment and detection by using Compound Discoverer 3.3 software, 3493 and 2573 features were obtained in all fresh-cut potato samples under the positive and negative mode, respectively.

TICs of quality control (QC) samples were also obtained from the positive and negative ionization modes. The overlapping peaks of QC samples in Figure 3A,B implied that the metabolites’ extraction and detection had good repeatability and the detection system had a high stability. QC samples were also used to ensure the precision and repeatability of data. All features with a coefficient of variation (CV) value < 30% in QC samples were retained to retain the stable features. All features with a detection rate > 50% in the QC samples were retained to keep the repeatable features. After filtering out features against the two rules, there were 517 repeatable and stable features in both modes retained for further statistical analyses.

### 2.3. Multivariate Statistical Analysis

The principal component analysis (PCA) model was established to evaluate any possible clustering of fresh-cut potato samples stored for different periods without providing prior sample classes information [32]. The first two principal components (PCs) explained the variations with PC1 and PC2 representing 23.5% and 17.5%, respectively. In PCA score plots (Figure 4A), samples were clearly clustered into four clusters: 0d, 2d, 4 to 6d and 8d. QC samples were tightly clustered, showing that the high quality of data acquisition was ensured. The distance was closer between 4d and 6d samples, meaning that the similarity of metabolite composition was higher. Conversely, with the storage time growing longer, the distance of these samples to 0d samples was getting farther correspondingly, indicating that metabolic variations were gradually significant.

The partial least squares discriminant analysis (PLS-DA) model was constructed to discriminate 0d and 8d fresh-cut potato samples with a pre-existent classification. The high values of R^2^Y = 0.997 and Q^2^ = 0.957 demonstrated the high explained variation and predictability of the model, respectively. The PLS-DA score plot (Figure 4B) showed obvious separation between 0d and 8d samples, indicating that the two groups could be completely distinguished based on constituent metabolites. The 200 times random permutation tests were performed to further validate the corresponding PLS-DA model [33]. R^2^ and Q^2^ values of the original PLS-DA model (the two rightmost points) were higher than those of the random permutation test model (all the points on the left), suggesting that the original model was not overfitted (Figure 4C).

### 2.4. Key Metabolites Analysis

The variable importance in projection (VIP) approach was applied to reveal the contribution of a metabolite to the PLS-DA model. Metabolites with VIP > 1 indicated that they best distinguished 0d and 8d sample groups [34]. Moreover, the *p*-value from the analysis of variance (ANOVA) and log_2_ fold change (FC) value were used to screen significantly changed metabolites. Metabolites with *p* < 0.05, absolute log_2_ FC > 1, and VIP >1 were considered to be differential metabolites between 0d and 8d fresh-cut potato samples. A total of 148 differential metabolites was visualized by a volcano plot (Figure 4D), including 125 up-regulated metabolites (red points) and 23 down-regulated (blue points) metabolites.

However, these differential metabolites were automatically detected from the raw data by the Compound Discoverer 3.3 software. The reliability of these metabolites should be confirmed by manual inspection. Firstly, duplicates needed to be removed. Secondly, the peak shape needed to be acceptable. More importantly, the structure of metabolites was identified using MS data search through ChemSpider database and MS/MS spectrum mapping through the mzCloud database. Finally, a total of 15 metabolites were identified as key metabolites responsible for the metabolic changes during the browning process of fresh-cut potatoes, comprising 4 amino acids and derivatives, 3 lipids and derivatives, 5 nucleotides and derivatives, 2 vitamins and 1 sugar. The parameters mentioned above of the 15 key metabolites were shown in Table 1.

Key metabolites were further validated using analytical standards under the same separation condition and similar data acquisition mode. Finally, ornithine, glutamic acid, glutathione, linolenic acid, phytosphingosine, cytidine, adenine and pyridoxine were perfectly confirmed by matching retention time and ion fragments. The ion fragments of S-adenosylmethionine, AMP and pyridoxamine were suppressed by neighboring compounds, which have similar m/z value and retention time but high abundance, so they were singly confirmed by matching retention time. 1-Methylhistidine, 12-OPDA, cytosine and maltose were single confirmed by ion fragments because the retention time was slightly shifted, which may be due to complex samples matrix.

### 2.5. Changes of Key Metabolites during Browning Process

Untargeted metabolomic workflow was used on the raw data to clarify correlative changes in key metabolites that occurred during browning process. The box plots of 15 key metabolites showing the variation in abundance were showed in Figure 4. During the 8 days of storage period, the levels of glutamic acid, pyridoxamine, pyridoxine, 1-methylhistidine, linolenic acid and glutathione showed a downward trend (Figure 5A–F), the levels of cytosine, ornithine, adenine, cytidine, maltose and S-adenosylmethionine showed an upward trend (Figure 5G–L), while the levels of 12-OPDA, phytosphingosine and AMP showed an upward trend but downward thereafter (Figure 5M–O). After integrating the potential mechanism of metabolic changes in the key metabolites, glutamic acid, linolenic acid, glutathione, adenine, 12-OPDA and AMP were used to explain metabolic causes of the browning of fresh-cut potatoes, while others still warranted further investigation due to the lack of relevant information.

#### 2.5.1. Amino Acids and Derivative

Glutamic acid is a wound signal molecule in plants, which can be transmitted to undamaged parts to induce defense responses when potatoes suffered mechanistic injury, such as fresh-cut treatment [35]. A recent study showed the content increase of glutamic acid decreased the browning sensitivity of potato and speculated that glutamic acid was directly involved in the enzymatic browning of fresh-cut potatoes [36]. Figure 4A showed that the level of glutamic acid significantly decreased as the storage period increased, implying that the defense response to browning gradually decreased.

In plants, reactive oxygen species (ROS) are maintained in a dynamic balance between production and scavenging; however, the balance is disrupted if the plant tissue suffers mechanical injury, leading to ROS accumulation and cell membrane damage [37]. Glutathione, as an antioxidant, is able to remove ROS via the ascorbate–glutathione cycle [38]. It was reported that glutathione effectively inhibited the browning of grape juice, since it could inhibit PPO activity [39]. In the present study, glutathione was significantly downregulated during storage, indicating that the antioxidant capacity of potato was weakened after cutting, which in turn made it more susceptible to browning (Figure 4). It was found that the browning of fresh-cut potato was accompanied by the decline of anti-browning substance content and activated oxygen scavenging ability.

#### 2.5.2. Lipids and Derivative

As shown in Figure 4, linolenic acid content continuously decreased during storage. 12-OPDA was present at a low intensity just after cutting, increasing rapidly from 0 to 6 days and then decreasing progressively (Figure 4). The membrane system of potato was mechanically damaged via fresh-cut processing, and phospholipids as an important component of membrane were broken down [40]. Fatty acids, mainly linoleic acid and linolenic acid, would be released from phospholipids, so the content of α-linolenic acid was highest at the initial stage of storage [41]. Then, α-linolenic acid was oxidized into 12-OPDA and finally metabolized to jasmonic acid through the octadecanoid pathway, leading to the consumption of linolenic acid and the accumulation of 12-OPDA [42,43]. Therefore, membrane disruption and fatty acid oxidation might occur during the browning process of fresh-cut potatoes. 

#### 2.5.3. Nucleotides and Derivative

According to the previous study, the development of browning in lotus seeds was accompanied with the decrease of adenosine triphosphate (ATP) levels, reflecting that the browning of lotus seeds might be related to energy shortage in the tissue [44]. In fruits and vegetables, energy plays a crucial role in maintaining membrane integrity and inhibiting oxidative damage [45]. Energy is provided by ATP decomposition in eukaryotes [46]. In this catabolic process, ATP was firstly degraded to adenosine diphosphate, then degraded to AMP through dephosphorylation reactions. AMP was dephosphorylated to adenosine, which was further degraded to form adenine [47]. From Figure 4, we observed that the AMP level in fresh-cut potatoes ascended gradually followed by a slight decline, with the peak value on 4 days of storage. Adenine content gradually increased along with the length of storage period (Figure 4). The above results suggested that an energy deficit might be associated with ATP breakdown, which further contributed to fresh-cut potato tissue browning.

### 2.6. Limitations

In this work, the untargeted metabolomics approach based on UHPLC-HRMS was performed to reveal key metabolites presenting in fresh-cut potato during the browning process. We acknowledge this study has certain limitations in the data acquisition method and metabolite identification. Hydrophilic interaction liquid chromatography (HILIC) is the most used chromatographic technique in untargeted metabolomics, which is suited to retain and separate polar compounds. However, it is a single separation technique used to separate metabolites in fresh-cut potatoes, so it can not cover all comprehensive metabolites in this complex biological sample. Thus, only polar metabolites in fresh-cut potatoes were addressed, but high-hydrophobic compounds may fail to be effectively retained. Metabolite identification is a complex step and has become a challenge in untargeted metabolomics research. Mass spectral library searching is the most common approach for compound identification in untargeted metabolomics at present, since it is fast and convenient. However, it is limited by relatively small experimental spectral libraries and the disruption from complicated sample matrix. Many metabolites were not identified because the metabolite identification is heavily dependent on online spectral libraries.

## 3. Materials and Methods

### 3.1. Chemical and Reagents

Analytical standards of glutamic acid, pyridoxamine, pyridoxine, 1-methylhistidine, linolenic acid, glutathione, cytosine, adenine, cytidine, maltose, S-adenosylmethionine, 12-OPDA, phytosphingosine, AMP and ornithine were purchased from MedChemExpress (Princeton, NJ, USA). All solvents and reagents were of high-performance liquid chromatography (HPLC) grade or higher. Methanol, ammonium formate, acetonitrile and formic acid were purchased from Sigma-Aldrich (St. Louis, MO, USA). Ultrapure water (at 18.2 MΩ·cm) was produced by Milli-Q system (Millipore Corp., Bedford, MA, USA).

### 3.2. Fresh-Cut Processing

Thirty fresh potato tubers with uniform size and free from mechanical damage were obtained from a local vegetable market in Zhengzhou, China. They were washed, peeled and cut into thin slices with thicknesses of 5 mm. Then, these potato slices were packed into polyethylene plastic bags and were stored in the refrigerator (4 °C) for 0, 2, 4, 6 and 8 days. The samples were assigned to five groups, named as 0d, 2d, 4d, 6d and 8d. Each sample group contains 6 biological replicates.

### 3.3. Color Measurement

The surface color of fresh-cut potato slices was recorded by CM−5 spectrophotometer (Konica Minolta, Co., Ltd., Tokyo, Japan) with D65 light source and 10° angle observers. Three color parameters of the CIELAB colorimetric system, L*, a* and b* were measured using the spectrophotometer [48]. Total color difference ΔE* was compared to the color of initial time, which was calculated as follows: $∆E*=\sqrt{{∆L*}^{2}+{∆a*}^{2}+{∆b*}^{2}}$ [49]. The visual appearance of slices was monitored by taking pictures.

### 3.4. Metabolite Extraction

For each storage period, fresh-cut potato samples were ground to powder in liquid nitrogen with a mortar, followed by lyophilization and storage at −80 °C. Each sample of 40 mg was extracted by 1 mL extraction solvent (methanol:water = 4:1, *v*/*v*). The mixture was then vortexed for 1 min, sonicated for 1 h in an ice-water bath and centrifuged at 13,000 rpm and 4 °C for 20 min. Finally, the supernatant was filtered through a 0.22 µm nylon syringe filter prior to UHPLC-HRMS analysis. QC samples were prepared by mixing equal volumes of each sample to evaluate the stability and reproducibility of analytes detected in the samples.

### 3.5. UHPLC-HRMS Acquisition

Metabolite extracts of fresh-cut potato samples were analyzed using a Dionex UltiMate 3000 UHPLC system coupled with a heated electrospray ionization (HESI) source and a Q-Exactive HRMS (Thermo Fisher Scientific, Waltham, MA, USA). A total of 1 µL of extract was separated on an ACQUITY UPLC BEH Amide column (2.1 × 100 mm, 1.7 µm) (Waters Corporation, Milford, MA, USA). The mobile phase was composed of (A) 10 mM ammonium formate and 0.125% formic acid in ultrapure water and (B) acetonitrile. The gradient was as follows: 0−1 min, 95−90% A; 1−4 min, 90−70% A; 4−11 min, 70% A; 11−13 min, 70−95% A; 13−15 min, 95% A. The constant flow rate was 0.25 mL/min and the column temperature was kept at 30 °C.

The ion source was operated in positive and negative ionization modes and various parameters were as follows: spray voltage was +3.8 kV for positive mode and −3.1 kV for negative mode, capillary temperature was 320 °C, heater temperature was 250 °C, sheath gas flow was 35 arbitrary units, auxiliary gas flow was 15 arbitrary units, sweep gas flow was 1 arbitrary units and S-lens RF level was 55 V. Data acquisition was carried out by combining the full scan MS and the dd-MS^2^. For the full MS mode, the resolution was 70, 000 and the m/z scan range was 70−1000. For the dd-MS^2^ mode, the resolution was 17,500 and the top 10 intense ions were fragmented by the stepped normalized collision energy at 20%, 40% and 60%.

### 3.6. Data Processing and Statistical Analysis

UHPLC-HRMS raw data was processed by Compound Discoverer 3.3 software (Thermo Fisher Scientific, Waltham, MA, USA). The untargeted metabolomics workflow was used for peaks alignment, peaks detection, features filtering and metabolites annotation. Peaks alignment parameters mainly included retention time (RT) and mass tolerances, which were set as 0.2 min and 5 ppm, respectively. Peaks detection was performed using signal-to-noise (S/N) at 1.5 and peak intensity thresholds at 10,000. All features with detection rate ≤ 50% and CV ≥ 30% in the QC samples were filtered out to reduce the number of poorly repeatable and unrobust features. Metabolites were putatively annotated using ChemSpider database by comparing theoretical and detected MS data and also mzCloud database by matching the referential and detected MS/MS spectrum [50].

The SIMCA 14.1 software (Umetrics, Umea, Sweden) was used for multivariate statistical analysis based on the metabolites’ peak areas. PCA was carried out to distinguish all fresh-cut potato samples based on metabolite differences. PLS-DA was performed to discriminate samples stored for 0 day and 8 days. Permutation tests (200 times) were used to evaluate the effectiveness of PLS-DA model [51]. The VIP values were calculated to select those metabolites having the highest discrimination potential in the PLS-DA models. Subsequently, the ANOVA and FC analysis were used as a univariate approach for selecting metabolites with significant changes in abundance. Differential metabolites with *p* < 0.05, absolute log_2_ FC > 1 and VIP > 1 were screened for metabolites identification [52].

### 3.7. Metabolites Confirmation

Analytical standards were used to confirm the metabolites using the previous UHPLC-HRMS conditions, except that data acquisition was carried out by parallel reaction monitoring (PRM) mode with the resolution at 17,500. Sample retention time and ion fragments were key confirmative indicators.

## 4. Conclusions

Metabolic profiling was analyzed to explore the metabolic changes of fresh-cut potato during the browning process. Untargeted metabolomics analysis revealed that browning was accompanied by changes in the small metabolites. The results suggested that 148 differential metabolites were screened between 0d and 8d fresh-cut potato samples. Fifteen key metabolites responsible for the browning process were putatively identified. Nine key metabolite levels were significantly increased, including cytosine, ornithine, adenine, cytidine, 12-OPDA, phytosphingosine, maltose, AMP and S-adenosylmethionine, whereas six key metabolite levels were significantly decreased, including glutamic acid, pyridoxamine, pyridoxine, 1-methylhistidine, linolenic acid and glutathione. The different variation trends of glutamic acid, linolenic acid, glutathione, adenine, 12-OPDA and AMP were related to the structural dissociation of the membrane, oxidation and reduction reaction and energy insufficiency. This work provides reliable evidence for compositional changes during the browning process of fresh-cut potatoes and a reference for further investigation into the mechanism of browning in other fresh-cut products.

## Figures and Tables

**Figure 1 molecules-28-03375-f001:**
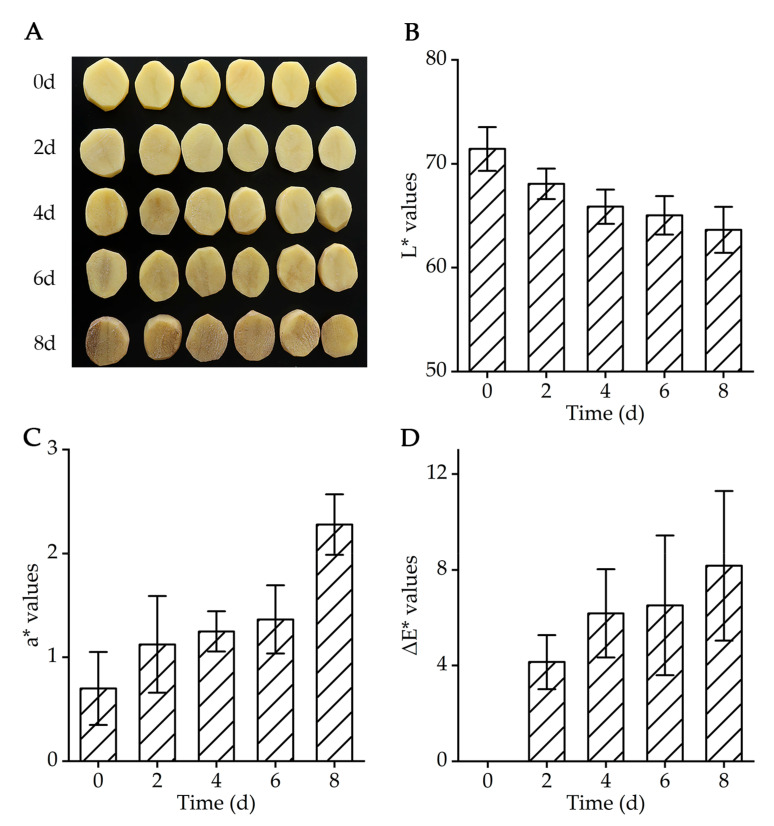
Changes in the color of fresh-cut potato slices stored at 4 °C for 8 days. (**A**) Photographs, (**B**) L* values, (**C**) a* values and (**D**) ΔE*. Data are expressed as the mean ± standard deviation (*n* = 6).

**Figure 2 molecules-28-03375-f002:**
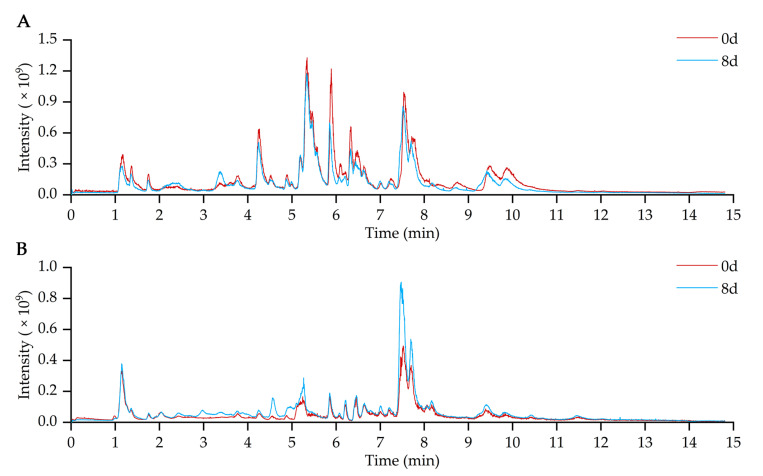
The typical total ion chromatograms of 0d and 8d fresh-cut potato samples acquired in (**A**) positive and (**B**) negative modes.

**Figure 3 molecules-28-03375-f003:**
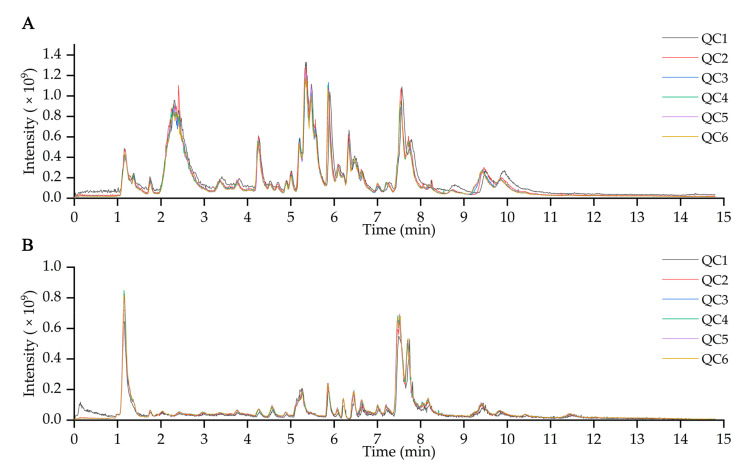
Total ion chromatograms of QC samples acquired in (**A**) positive and (**B**) negative modes.

**Figure 4 molecules-28-03375-f004:**
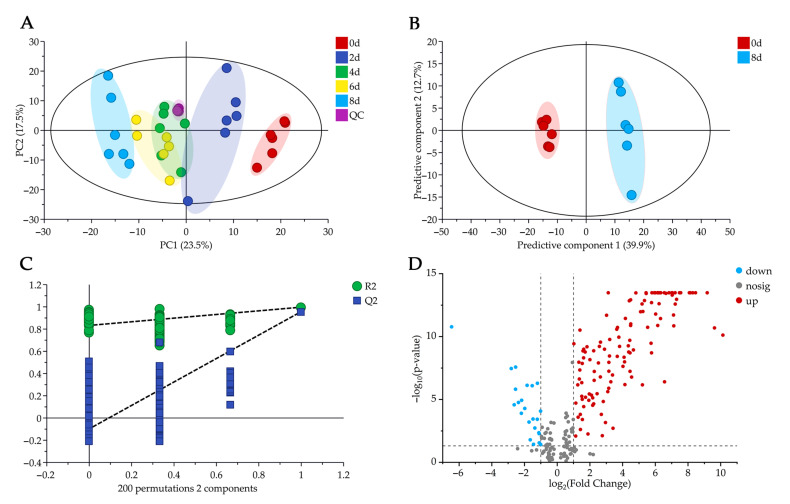
Statistical analysis of fresh-cut potato samples during different storage periods. (**A**) PCA score plot showing clustering for 0d, 2d, 4d, 6d, 8d and QC samples, (**B**) PLS-DA score plot for 0d versus 8d samples, (**C**) 200 permutation tests plot for the PLS-DA model and (**D**) volcano plot of significantly changed metabolites for 8d versus 0d samples.

**Figure 5 molecules-28-03375-f005:**
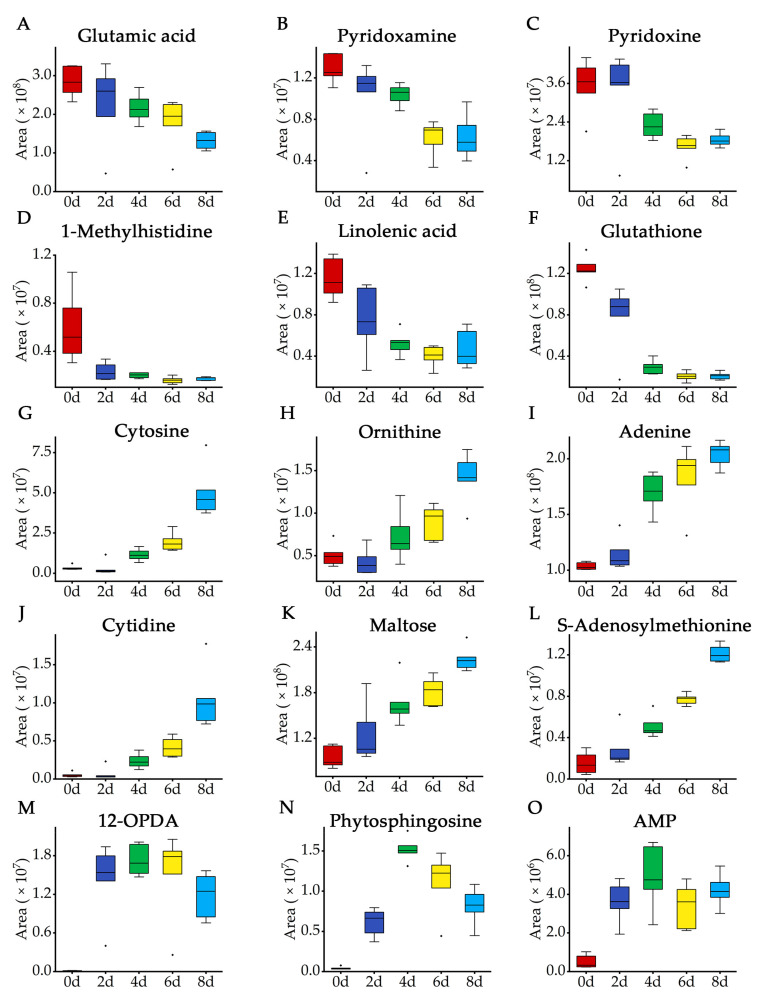
Box plots showing abundance variations of 15 key metabolites in fresh-cut potato stored from 0 to 8 days. In the box plot, center lines show the medians, box limits indicate the 25th and 75th percentiles, whiskers extend 1.5 times the interquartile range from the 25th and 75th percentiles, outliers are represented by dots. *n* = 6 sample points.

**Table 1 molecules-28-03375-t001:** The list of 15 key metabolites identified in the 0d compared to 8d fresh-cut potato samples.

Name	Formula	Molecular Weight (Da)	Mass Error (ppm)	Retention Time (min)	VIP Score (0d/8d)	*p*-Value (0d/8d)	Log_2_ FC (0d/8d)
Amino acids and derivatives							
Ornithine	C_5_H_12_N_2_O_2_	132.0896	−1.90	10.12	1.46	9.81 × 10^−6^	1.53
Glutamic acid	C_5_H_9_NO_4_	147.0527	−3.04	7.61	1.49	2.86 × 10^−2^	−1.10
1-Methylhistidine	C_7_H_11_N_3_O_2_	169.0847	−2.46	9.33	1.19	8.16 × 10^−7^	−1.53
Glutathione	C_10_H_17_N_3_O_6_S	307.0829	−2.91	7.93	1.57	2.70 × 10^−8^	−2.54
Lipids and derivatives							
Linolenic acid	C_18_H_30_O_2_	278.2238	−2.89	1.27	1.43	3.60 × 10^−4^	−1.48
12-OPDA	C_18_H_28_O_3_	292.2028	−3.67	2.39	1.49	1.10 × 10^−13^	7.30
Phytosphingosine	C_18_H_39_NO_3_	317.2919	−3.59	4.85	1.49	1.19 × 10^−13^	4.49
Nucleotides and derivatives							
Cytosine	C_4_H_5_N_3_O	111.0432	−0.47	5.85	1.46	2.32 × 10^−8^	4.02
Adenine	C_5_H_5_N_5_	135.0541	−2.71	3.37	1.57	3.70 × 10^−10^	1.02
Cytidine	C_9_H_13_N_3_O_5_	243.0846	−3.63	5.86	1.42	1.70 × 10^−9^	4.37
AMP	C_10_H_14_N_5_O_7_P	347.0621	−2.86	7.92	1.52	3.75 × 10^−9^	3.67
S-Adenosylmethionine	C_15_H_22_N_6_O_5_S	398.1363	−2.26	10.76	1.57	1.12 × 10^−8^	3.16
Vitamins							
Pyridoxamine	C_8_H_12_N_2_O_2_	168.0894	−2.66	7.25	1.43	5.22 × 10^−3^	−1.11
Pyridoxine	C_8_H_11_NO_3_	169.0734	−2.97	3.89	1.35	3.76 × 10^−2^	−1.01
Sugar							
Maltose	C_12_H_22_O_11_	342.1151	−3.26	7.46	1.56	1.10 × 10^−8^	1.32

## Data Availability

The data that support the findings of this study are available from the corresponding author upon reasonable request.

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
