# Peer review of "Untargeted Metabolomics Using UHPLC-HRMS Reveals Metabolic Changes of Fresh-Cut Potato during Browning Process"

_molecules, 2023, doi:10.3390/molecules28083375_

Round 1

Reviewer 1 Report

I have enjoyed reading this manuscript. Please see below for my comments.

Specific comments: 

1. "... biological systems and identify the metabolites which could play the most relevant role in various processes [13,14]" - as this is referring to biological systems rather than the applications in food safety, suggest to replace ref 14 here with a separate ref (citation: pubmed.ncbi.nlm.nih.gov/30056340).

2. "... stored in the refrigerator (4 °C) for 0, 2, 4, 6, and 8 days" - it is unclear why this specific condition and interval was chosen by the authors. In the real world, potatoes are usually not refrigerated and the browning happens rather quickly when exposed to air at ambient temperature. 

3. Were the potato slices taken from the same potato or it was ensured that the slices would be from different ones? The metabolomics standards initiative (MSI) recommends a minimum of 5 biological replicates in their minimum reporting standards, but of course the true number required depends heavily on the intrinsic variation in the biological samples as well as the magnitude of the observed perturbation, all factors incorporated into power analysis.

4. "517 repeatable and stable 90 features in both modes" - please try not to begin a sentence with a number.

5. Besides mention of the sample replicates, the nature of replication in the experimental design is unclear, and the assessment of uncertainty in the reported measurement is absent or unclear.

6. From sample acquisition, to sample extraction and chromatographic selection, one can heavily bias the metabolites resolved in an untargeted metabolomics study, necessitating careful scrutiny and validation of each facet of the experiment. Suggest authors include some discussion of study limitations and critique of their own methods.

Author Response

Comment 1: "... biological systems and identify the metabolites which could play the most relevant role in various processes [13,14]" - as this is referring to biological systems rather than the applications in food safety, suggest to replace ref 14 here with a separate ref (citation: pubmed.ncbi.nlm.nih.gov/30056340).

Response 1: We fully agree with the comment and we have replaced the reference 14 according to your suggestion. This reference is now included in the revised manuscript.

“14. Wang, P.; Ng, Q.X.; Zhang, H.; Zhang, B.; Ong, C.N.; He, Y. Metabolite changes behind faster growth and less reproduction of Daphnia similis exposed to low-dose silver nanoparticles. Ecotoxicol. Environ. Saf. 2018, 163, 266-273, doi:10.1016/j.ecoenv.2018.07.080.”

Comment 2:"... stored in the refrigerator (4 °C) for 0, 2, 4, 6, and 8 days" - it is unclear why this specific condition and interval was chosen by the authors. In the real world, potatoes are usually not refrigerated and the browning happens rather quickly when exposed to air at ambient temperature.

Response 2: Thank you for your kind question. It is really true as reviewer considered that the browning happens rather quickly when exposed to air at ambient temperature. So Fresh-cut potato slices were placed in the refrigerator (4 °C) to slow down the browning development and more clearly observe surface color changes. As the storage time increases, the surface color gradually tended to brown, and it had become very obvious and unacceptable on the eighth day, so experiments were terminated at this time. Since the samples showed a clearer color difference every two days, the samples were collected from five different browning stages for further analysis.

Comment 3: Were the potato slices taken from the same potato or it was ensured that the slices would be from different ones? The metabolomics standards initiative (MSI) recommends a minimum of 5 biological replicates in their minimum reporting standards, but of course the true number required depends heavily on the intrinsic variation in the biological samples as well as the magnitude of the observed perturbation, all factors incorporated into power analysis.

Response 3: Thank you for your kind question. We are very sorry for the question in this manuscript and inconvenience it caused in your understanding. Each potato slice was taken from an independent potato. A total of 30 potato tubers were collected and divided into five groups, named 0d, 2d, 4d, 6d and 8d. Each sample group contains 6 biological replicates according to the metabolomics standards initiative (MSI). We have corrected “Section 3.2. Fresh-cut processing” accordingly in the revised manuscript.

Comment 4: "517 repeatable and stable 90 features in both modes" - please try not to begin a sentence with a number.

Response 4: We thank you for pointing out this issue and we have thoroughly checked and corrected the description accordingly. The sentence “517 repeatable and stable features in both modes with detection rate ≥ 50% and coefficient of variation (CV) ≤ 30% in the QC samples were retained for further statistical analyses.” now reads “After filtering out features against the two rules, there were 517 repeatable and stable features in both modes retained for further statistical analyses.”

Comment 5: Besides mention of the sample replicates, the nature of replication in the experimental design is unclear, and the assessment of uncertainty in the reported measurement is absent or unclear.

Response 5: Thank you for pointing out this problem in manuscript. We have added the content of data quality evaluation on paragraph 2, section 2.2 in the revised manuscript. We have also included the content here for your perusal:

TICs of quality control (QC) samples were also obtained from the positive and nega-tive ionization modes. The overlapping peaks of QC samples in Figure 3A and B implied that metabolites extraction and detection had good repeatability and detection system had high stability. QC samples were also used to ensure the precision and repeatability of data. All features with a coefficient of variation (CV) value < 30% in QC samples were retained to retain the stable features. And all features with a detection rate > 50% in QC samples were retained to keep the repeatable features. After filtering out features against the two rules, there were 517 repeatable and stable features in both modes retained for further statistical analyses.

Comment 6: From sample acquisition, to sample extraction and chromatographic selection, one can heavily bias the metabolites resolved in an untargeted metabolomics study, necessitating careful scrutiny and validation of each facet of the experiment. Suggest authors include some discussion of study limitations and critique of their own methods.

Response 6: Thank you for your kind suggestions. We have added the suggested content to the revised manuscript on section 2.6. We have also included the content here for your perusal:

2.6 Limitations

In this work, untargeted metabolomics approach based on UHPLC-HRMS was per-formed to reveal key metabolites presenting in fresh-cut potato during browning process. We acknowledge this study has certain limitations in data acquisition method and metabolite identification. Hydrophilic interaction liquid chromatography (HILIC) is the most used chromatographic techniques in untargeted metabolomics, which is suited to retain and separate polar compounds. While it was it was single separation technique used to separate metabolites in fresh-cut potatoes, so it can not cover all comprehensive metabo-lites in this complex biological sample. Thus, only polar metabolites in fresh-cut potatoes was addressed, but high-hydrophobic compounds may fail to be effectively retained. Metabolite identification is a complex step and has become a challenge in untargeted metabolomics researches. Mass spectral libraries searching is the most common approach for compounds identification in untargeted metabolomics at present, since it is fast and convenient. However, it is limited by relatively small experimental spectral libraries and the disruption from complicated sample matrix. There are many metabolites failed to identify because the metabolite identification are heavy dependency on online spectral libraries.

Reviewer 2 Report

The manuscript entitled "Untargeted Metabolomics Using UPHLC-HRMS Reveals Metabolic Changes of Fresh-Cut Potato during Browning Process" aims to unravel the metabolic changes occurring after potato cutting. An untargeted metabolomics approach based on UHPLC-HRMS was performed to determine the metabolites presenting in fresh-cut potatoes during storage. Differences in metabolites were assessed using Compound Discoverer 3.3 software and statistical analysis. The results discovered the key metabolites responsible for the browning process and elucidated the causes of metabolic changes in relation to the browning process. The manuscript is very interesting. It is well organized in general, although it is a bit short. My specific comments are given below.

The abstract is suitable.

The introduction is short and not very informative. It should be improved in order to provide more relevant details, such as possible reasons for potato browning and metabolomics uses.

Material and methods are described in detail.

The results are well presented. Still, the discussion could be more extensive (Section 2.5.). In this regard, it would be beneficial if one of the metabolites was confirmed with the analysis of analytical standards and its concentration was determined. It could be done for metabolites that are present in the highest concentrations.

The conclusion is supported by the results. 

The literature is up-to-date.

Author Response

Comment 1: The abstract is suitable.

Response 1: Thank you so much for your careful check.

Comment 2: The introduction is short and not very informative. It should be improved in order to provide more relevant details, such as possible reasons for potato browning and metabolomics uses.

Response 2: Thank you for your kind suggestions. We have added the suggested content to paragraph 2, page 1 for possible reasons for potato browning and paragraph 3, page 2 for metabolomics uses in the revised manuscript. We have also included the content here for your perusal:

In normal plant tissues, according to phenol-phenolase regionalized distribution theory, phenolic substances and polyphenol oxidase (PPO) were mainly found in vacuoles and plastids, respectively [5]. However, once cell integrity was destroyed by fresh-cut processing, PPO came into contact with phenolic substrate, and they were exposed to air simultaneously, which induced the occurrence of enzymatic oxidation reactions as three elements of oxygen, substrate and enzyme coexisted. So after cutting, phenolic substrates were catalyzed to quinones, which could polymerize rapidly and react with amino or sulphydryl of protein and sugars through a non-enzymatic reaction into dark brown pigments with adverse effects on the appearance and consumer preference of the fresh-cut products.

For example, it was used for reveal metabolic changes in electron beam irradiated shrimp meat during refrigerated storage, late blight-infected tomato at different times post infection and fresh-cut Chinese water chestnuts at different yellowing stages, biochemical changes in Laowo ham during ripening and black garlic during aging process, chemical constituents changes in instant dark tea during fermentation by Eurotium cristatum.

Comment 3: Material and methods are described in detail.

Response 3: Thank you so much for your careful check.

Comment 4: The results are well presented. Still, the discussion could be more extensive (Section 2.5.). In this regard, it would be beneficial if one of the metabolites was confirmed with the analysis of analytical standards and its concentration was determined. It could be done for metabolites that are present in the highest concentrations.

Response 4: Thank for your kind commons and we have further confirmed the key metabolites using available chemical standards. We are sorry for the lack of quantitative result because limited time. We have added the discussion and analytical method to the revised manuscript on paragraph 3, section 2.4 and section 3.7, respectively. We have also included the content here for your perusal:

Key metabolites were further validated using analytical standards under same separation condition and similar data acquisition mode. Finally ornithine, glutamic acid, glutathione, linolenic acid, phytosphingosine, cytidine, adenine, and pyridoxine were per-fectly confirmed by matching retention time and ion fragments. The ion fragments of S-adenosylmethionine, AMP and pyridoxamine were supressed by neighboring compounds which have similar m/z value and retention time but high abundance, so they were singlely confirmed by matching retention time. 1-Methylhistidine, 12-OPDA, cytosine and maltose were single confirmed by ion fragments, because the retention time were slightly shifted, which may be due to complex samples matrix.

3.7 Metabolites confirmation

Analytical standards were used to confirm the metabolites using the previous UHPLC-HRMS conditions, except that data acquisition was carried out by parallel reac-tion monitoring (PRM) mode with the resolution at 17,500. Sample retention time and ion fragments were key confirmative indicators.

Comment 5: The conclusion is supported by the results.

Response 5: Thank you so much for your careful check.

Comment 6: The literature is up-to-date.

Response 6: Thank you so much for your careful check.

Round 2

Reviewer 1 Report

Thank you for the revisions.